# Facile Fabrication of Thin-Bottom Round-Well Plates Using the Deformation of PDMS Molds and Their Application for Single-Cell PCR

**DOI:** 10.3390/mi11080748

**Published:** 2020-07-31

**Authors:** Shinya Yamahira, Yuji Heike

**Affiliations:** 1Center for Medical Sciences of St. Luke’s International University, 3-6-2, Tsukiji, Chuo-ku, Tokyo 104-0045, Japan; heiyuji@luke.ac.jp; 2Graduate School of Public Health and Hospital at St Luke’s International University, 9-1, Akashi-Cho, Chuo-ku, Tokyo 104-8560, Japan

**Keywords:** polymethylsiloxane mold, elastic mold, microfabrication, heat-press, thin wall, single-cell polymerase chain reaction, multi-well plate

## Abstract

Recently, microdevices made of resins have been strongly supporting cell analysis in a range of fields, from fundamental life science research to medical applications. Many microdevices are fabricated by molding resin to a mold made precisely from rigid materials. However, because dimensional errors in the mold are also accurately printed to the products, the accuracy of the product is limited to less than the accuracy of the rigid mold. Therefore, we hypothesized that if dimensional errors could be self-corrected by elastic molds, microdevices could be facilely fabricated with precision beyond that of molds. In this paper, we report a novel processing strategy in which an elastic mold made of polymethylsiloxane (PDMS) deforms to compensate for the dimensional error on the products. By heat-press molding a polycarbonate plate using a mold that has 384 PDMS convexes with a large dimensional error of height of ± 15.6 µm in standard deviation, a 384-round-well plate with a bottom thickness 13.3 ± 2.3 µm (*n* = 384) was easily fabricated. Finally, single-cell observation and polymerase chain reactions (PCRs) demonstrated the application of the products made by elastic PDMS molds. Therefore, this processing method is a promising strategy for facile, low-cost, and higher precision microfabrication.

## 1. Introduction

Resin processing technologies based on rigid molds are some of the most essential technologies to supply tools that support life sciences and healthcare. Consumable goods, such as test tubes, culture dishes, and syringes, have long been mass-produced by applying a heat-softened plastic or curable resin to a metal mold, using molding technologies such as injection molding [1], vacuum/pressure forming [2], and heat-press molding [3]. Currently, resin products finer and more precise than conventional ones are contributing to the development of fundamental life science research and pharmaceutical and medical applications, such as cell analysis devices. For example, these so-called microdevices, such as microwell arrays for single-cell manipulation [4,5], microfluidic channels for rapid and high-throughput polymerase chain reaction (PCR) [6,7], and drug testing [8,9], and point-of-care testing (POCT) devices for blood cell analysis [10], have been studied extensively. Some of these are already commercially available as high-value-added life science/medical products for cell analysis [11,12]. In particular, microfabricated chips have become essential resin products for controlling droplets and cells for advanced gene analysis technologies, such as digital PCR [13,14] and single-cell PCR [15,16].

In microfabrication technologies, molds made from rigid materials to promote dimensional stability are key items that are also used in traditional resin processing technologies. These molds are precisely fabricated using high-precision machining [17,18], lithography [19,20], laser processing [21], and electroforming [22]. However, these complex technologies result in higher prices for molds, and it becomes a problem in the development of microdevices. In turn, the microfabricated products are too expensive to be consumables, so researchers and medical staff draw limited benefits from such powerful cell analysis products. Therefore, a mold for microfabrication that can be easily prepared without high-precision processing is especially desired.

It is difficult to fabricate microstructures using rigid molds with high dimensional errors. This is because the products are created as complete transcripts of the mold, including its errors. For example, when fabricating a structure of several tens of micrometers in thickness, only a few micrometers of error on the mold are allowed. In other words, highly precise and expensive molds are required for microfabrication, because current molds are too rigid to compensate for their own dimensional errors and non-uniformity in the molding process [23]. Therefore, we hypothesized that the errors could be self-corrected by utilizing the deformation of elastic polymethylsiloxane (PDMS) molds. PDMS molds have been reported to have easy handling and mold release. The molding processes were designed to prevent excessive pressure on the molds to avoid mold deformation [24,25,26,27]. As a few studies that take advantage of the softness of PDMS molds, soft mold-based hot embossing on non-planar surfaces was reported. A micro-lens array was constructed on a plastic lens by fitting a membrane-like PDMS mold to the surface of the lens, by applying optimized air pressure [28]. However, in this study, we fabricated a microstructure in a product with small errors by deforming the PDMS mold, to compensate for its own dimensional errors by applying excessive pressure. Thin-walled bottoms of wells in a 384 round-well plate were molded as a model for microfabrication. The mold enabled the fabrication of well bottoms of around 10 μm in thickness, with a standard deviation of a few micrometers for each well, which are seamlessly connected to the thick-walled parts of the plate with curves. Finally, single-cell microscopy observations and single-cell PCR were performed as applications using the plate.

## 2. Principle and Mold Design

### 2.1. Principle for Fabrication of Thin Well Bottoms with Small Dimensional Error

The newly developed 384-round-well plate was simply fabricated by pressing polycarbonate plates between a convex PDMS mold and the glass plate at high temperatures above the glass transition point (Figure 1a). This PDMS mold was not precisely fabricated and had dimensional errors of tens of micrometers in the height between each convex. Despite the non-uniformity in pressure from each convex caused by a dimensional error of the mold on the resin, the thin bottoms of the wells were fabricated with a small error in thickness over a large area, including the 384 wells, using two principles. The first principle is that the dispersion in the pressure is restrained by the deformation of the PDMS mold. The second principle is that excess pressure is needed to make thin resins even thinner (Figure 1b). To make the resin thinner, it is necessary to extrude the resin from between the tip of the mold convex and the glass plate. The thinner the resin is, and the larger the contact area between the resin and the convex and the lower glass plate, the more pressure is required for the resin to flow, because of large shear stress and pressure loss. This second principle enhances the correction effect for the resin thickness on the first principle, and the dimensional error in the thickness of each thin well bottom is strongly suppressed. 

### 2.2. Design of PDMS Mold

The wells were designed to have a rounded shape, to collect the seeded cells on a relatively flat bottom at the center of the well. Therefore, cells seeded by a limiting dilution method allowed it to be easy to check whether each well contained single cells or not via microscopic observation. The multi-well plate was made of heat-resistant polycarbonate for use in PCR.

We planned to press a 1-mm-thick polycarbonate plate with a mold with a 2-mm-diameter PDMS convex to prepare a well. If the PDMS mold was not deformed, the height of the convex was calculated to be 1.122 mm, to fill the mold with softened polycarbonate, according to Figure 1c. By setting the convex height to 1.222 mm which ensures PDMS deformation, the thin well bottom surface was created by pressing with a strong but self-compensated pressure, using the PDMS convex tip. 

## 3. Materials and Methods 

### 3.1. Construction of the PDMS Mold

To construct the convex PDMS mold, we employed an air-pressure-assisted method to fabricate the lens array reported in some studies [29,30]. Figure 2 shows the construction process for the convex PDMS mold. The PDMS convex was fabricated by curing PDMS prepolymer on a PDMS membrane deformed into a concave shape, using negative pressure. First, 5 g of PDMS prepolymer (Sylgard 184, Dow Corning, MI, USA) composed with a 15:1 base to catalyst ratio was poured onto the polyethylene naphthalate (PEN) film (TEONEX Q51-25, Teijin Ltd., Tokyo, Japan). The 15:1 base to catalyst ratio was employed to obtain softer PDMS membrane than the manufacturer’s recommended ratio of 10:1 [31]. Air bubbles were removed by vacuum, and then the film was spin-coated at 1200 rpm for 2 min. After curing by heating at 125 °C for 20 min, the PDMS membrane was adhered to an acrylic plate using double-sided tape for silicone rubber. This acrylic plate had 2-mm-diameter through-holes drilled in the position, corresponding to the well of the 384-well microplate. The PEN film acting as support for the PDMS membrane was peeled off from the PDMS membrane on the acrylic plate. By using a silicone adhesive as a seal, the acrylic plate with the PDMS membrane was placed on top of the aluminum box, with a side hole for decompression. At this time, heat sinks for CPU were put in the aluminum box, to ensure the acrylic plate was not bent by negative pressure. This framework for the deformation of the PDMS membrane was used to investigate the relationship between the decompression force and the deformation of the PDMS membrane found in Section 4.1. Four strips of polystyrene (5 mm wide, 100 mm long, and 0.5 mm thick) were placed on the four sides of the acrylic plate as spacers and fences, to pour the PDMS prepolymer. Approximately 15 mL of PDMS prepolymer (10:1 base to catalyst ratio) was poured onto the PDMS membrane and spread to cover 384 wells, by blowing with nitrogen gas. After the PDMS prepolymer was degassed in a desiccator at reduced pressure, the aluminum box was connected to the regulator (PSR-01, As One, Osaka, Japan) and diaphragm pump (DA-30S, ULVAC, Kanagawa, Japan), using a polyurethane tube in its side hole, and depressurized at −40 kPa gauge pressure. The PDMS prepolymer was then covered with heat-resistant glass (150 mm wide, 150 mm long, and 3 mm thick) (Tempax, Schott, Mainz, Germany) and cured at 60 °C for 3 h, by using a heating plate of a heat-press machine (HC300-01, As One). The acrylic plate and spacers were removed from the PDMS mold on the heat-resistant glass after cooling. Finally, the PDMS mold was post-cured by heating at 180 °C for 3 h.

### 3.2. Fabrication of Multi-Well Plate

A polycarbonate plate (10 cm wide, 14 cm long, and 1 mm thick) was placed on a lower glass plate, which was prepared by spin-coating glass with PDMS as the release agent at 3000 rpm, for 3 min before being cured at 125 °C for 20 min. Then, the PDMS mold was placed on the polycarbonate plate. These were set in a heat-press machine pre-warmed to 170 °C, preheated for 8 min without the press, and then pressed at 900 ± 50 kg for 4 min at the same temperature. The press machine was water cooled to 90 °C, and the polycarbonate product was recovered from the PDMS mold. The polycarbonate product was sonicated at 23 kHz for 15 min three times with ethanol at room temperature, rinsed once with water, and air-dried on a clean bench.

### 3.3. Characterization of the PDMS Membrane, Mold, and Polycarbonate Product

The shapes of the PDMS membrane, mold, and polycarbonate products were observed in three dimensions by the Z-stack mode of the laser scanning confocal microscope (SP-8, Leica Microsystems Japan, Tokyo, Japan). First, the relationship between the decompression force and the deformation of the PDMS membrane was investigated to create a PDMS convex with the desired shape. The sag height of the deformed PDMS membrane was calculated from the vertical position of the PDMS membrane at the center and the outside of the hole. By detecting the laser reflection from the surface, it was possible to estimate the three-dimensional position of the unstained PDMS membrane by using confocal microscopy (ex/em = 488/483–493 nm, 5× objective lens, 14.4-μm step size z-stack) [32]. The thickness of the PDMS membrane was also measured by detecting the laser reflections from each side of the membrane using a 10× objective lens with a 2.02 μm step size z-stack.

Next, the dimensions of the PDMS convexes and polycarbonate wells were measured. In this experiment, the height of the convex (5× objective lens, 14.4-μm step size z-stack) and the thickness of the bottom of the well (10× objective lens, 1.34-μm step size z-stack) were determined, using the same method as described above. In addition, the three-dimensional shapes of the convex mold and the wells of the product were compared. It was difficult to observe the peripheral walls of both the convex and the well using the above method, because the detectable reflected laser was greatly weakened on the surface not vertical to the optical axis. Therefore, each surface was coated with 1% w/v bovine serum albumin–tetramethylrhodamine (BSA-TAMRA) (A2289, Sigma-Aldrich, St. Louis, MO, USA)/PBS, after which the three-dimensional shape was observed by conventional fluorescence observation (ex/em = 552/560–600 nm, 10× objective lens, 2.39-μm step size z-stack).

### 3.4. Monitoring of PCR Temperature

A small amount of silicone adhesive was applied to the wire of 0.076-mm-diameter K-type thermocouple (5SC-TT-K-40-36, Omega Engineering, Norwalk, CT, USA), to prevent leakage of the PCR mixture. This K-type thermocouple was inserted into the well without the part of thermal sensor contacting the surface of the well. The well was filled with 3 μL of the PCR mixture, and sealed with the adhesive PCR plate seal (ThermalSeal RTS, EXCEL Scientific, Victorville, CA, USA) over the thermocouple. The seal was completely bonded to the plate by being pressed at 900 kg for 10 min, using a press machine at room temperature. An 8-Channel Thermocouple Data Logger (TC-08, PICO technology, St Neots, England) was used to record the temperature every 100 ms during thermal cycling. 

### 3.5. Strain and Cell Culture

A murine pro-B cell line, Ba/F3, was obtained from Riken RBC (Tsukuba, Japan) and cultured in an RPMI 1640 medium (Thermo Fisher, Waltham, MA, USA) containing 10% fetal bovine serum (FBS), 100 U/mL penicillin, 100 µg/mL streptomycin, and 1 ng/mL IL-3 at 37 °C, in a humidified atmosphere containing 5% CO_2_ and 95% air. 

Ba/F3 cells stably expressing Kusabira Orange (KO) were prepared by conventional transfection with electroporation and cloning by limited dilution. Briefly, 40 µg of KO expression plasmids (AM-V0140, MBL, Tokyo, Japan) was mixed with 5 × 10^6^ cells of Ba/F3 cells in 200 µL Optimem (Thermo Fisher), transferred into a 0.2-cm-gap cuvette, and electroporated by delivering one exponential decay pulse of 140 V and 1000 µF, using the Gene Pulser Xcell Electroporation System with the CE Module (Bio-Rad, Hercules, CA, USA). Transfected cells were returned and cultured in RPMI 1640/10% FBS containing 1 ng/mL IL-3 for 18 h, and selected on the culture media supplemented with 1 mg/mL G418 for 1 month. Stably expressing KO cells were established by cloning with limited dilution in a 96-well plate.

### 3.6. Single-Cell PCR

PCR was performed using TaqMan Fast Advanced Master Mix (Thermo Fisher), in a Mastercycler Nexus flat (Eppendorf, Hamburg, Germany). The sequences of the primers and probes for the KO genes were as follows: forward primer, 5′-ATGGGCATGAGTTCACAATCG-3′; reverse primer, 5′-CGCAGTGTCATCTCCTGATGTC-3′; and probe, FAM-CAGGCAGACCTTACGAG-NFQ-MGB. PCR was carried in our 384-well plate with 3 μL of the PCR mixture per well containing 1.5 μL of TaqMan Fast Advanced Master Mix, 0.006 µL of 100 µM of each primer solution, at the final concentration of 0.2 µM each, 0.06 µL of 10.0 µM of Taqman probe solution at the final concentration of 0.2 µM, 1.128 µL of UltraPure DNase/RNase-Free Distilled Water (Thermo Fisher), and 1 or 10 cells/0.3 µL of Ba/F3 (KO) cell/PBS supplemented with 1% w/v BSA (A8806, Sigma-Aldrich) solution. BSA was supplemented at the final concentration of 0.1% w/v, to prevent the absorption of cells, nucleic acids, and enzymes into the wall of the wells [33,34]. PCR mixtures without cells served as the background for reporter signals. The plate was sealed with adhesive PCR plate seal. Cells in each well were simply counted by microscopic observations under bright-field and fluorescence, with excitation at 488 nm and emission at 500–550 nm, using a 10× objective lens with a 34.1-µm optical slice. Cells were lysed by this single freeze–thaw cycle (stored at −80 °C and thawed at room temperature for 2 h). The plate was installed into the thermal cycler, covered with an additional 384-well plate turned upside down, and held with a 96-well PCR plate and a heated lid from above. The target temperature for this PCR reaction was 20 s at 95 °C for initial denaturation, 45 cycles of 10 s at 95 °C for denaturing, 30 s at 62 °C for annealing and extension, followed by hold at 4 °C. The temperature of the heated lid was set at 78 °C. PCR in tube was carried with 20 μL of the PCR mixture per well containing 10 cells/3 µL of Ba/F3 (KO) cell/PBS supplemented with 1% w/v BSA, and the other reagents at the same final concentration, as described above.

### 3.7. Evaluation of PCR

The 6-carboxyfluorescein (FAM) (ex/em = 460/>515 nm) and 6-carboxy-X-rhodamine (ROX) (ex/em = 520/585–625 nm) fluorescence intensities of each well were measured by using ImageQuant LAS 4000 (GE Healthcare, Chicago, IL, USA). The amplification of the target gene on the PCR was represented as the background-corrected normalized fluorescent reporter signal (ΔRn), which was calculated as follows. The FAM fluorescence intensity from each well was normalized by dividing by the ROX fluorescence intensity from the same well, to obtain a ratio defined as the Rn. Notably, ΔRn is the Rn in each well minus the averaged Rn of six background wells in which the PCR mixtures did not contain cells. These quantitative image analyses were performed by ImageQuant TL (GE Healthcare).

## 4. Results and Discussion

### 4.1. Characterization of the Mold

By conducting three-dimensional observations of the laser reflection from the surface of the PDMS membrane, the decompression degree at which the height of the convex was close to 1.222 mm was inspected (Figure 3a). As shown in Figure 3b, a 1.23 mm sag height of the convex was accomplished with a −40 kPa gauge pressure. Even with these mild decompressions, the PDMS membrane (thickness = 33.0 ± 1.0 µm, *n* = 4) composed with a 15:1 base to catalyst ratio allowed preparation of the convex, with a height approximately up to 1.3 times the radius. The controllable range of the aspect ratio of the convex is adjustable, depending on the catalyst concentration, thickness of the membrane, and the type of PDMS used. As shown in Figure 3c, a convex PDMS mold was prepared without any defects on the convexes. Thin silicone frame (0.5-mm thickness) was attached to the peripheral of the PDMS molds, to flatten the peripheral of the 384-well plate (Figure 3c, left, indicated by red dotted line). By measuring the heights of 16 rows and 24 columns of the convexes, average height of 1200 ± 15.6 µm (*n* = 384) and maximum difference of 86.5 µm between convexes were found to characterize the molds (Figure 3d). The average height indicated that the molds were fabricated with a shape close to the design. However, the standard deviation and the maximum difference in height were greatly over 10 µm. If this were a rigid mold, these characteristics of the molds would be too non-uniform for molding well bottoms of a few tens of µm in thickness; wells with excessively thick bottoms and wells with a through-hole on the bottom would be produced.

### 4.2. Characterization of the Multi-Well Plate

By observing the polycarbonate product of the present process, we confirmed that a 384-round-well plate with a thin bottom was fabricated with no breakage or defects on the bottom of any of the wells (Figure 4a,b). Comparing the cross-section of the mold convex with that of the corresponding well by observing their surfaces coated with a fluorescent dye, the wells had a slightly compressed shape in the convex mold (Figure 4c). The bottom thickness of the 384-well plate was also measured at 13.3 ± 2.3 µm (*n* = 384 wells, Figure 4d). Additionally, the maximum difference in bottom thickness between these wells was 10.7 µm (20.1 µm maximum thickness and 9.4 µm minimum thickness). This result proved that elastic molds can be used to construct a thin structure of a few tens of micrometers thick, with a standard deviation of a few micrometers, even in an area as large as 78 cm^2^. This mold reduced dimensional errors in approximately one-seventh in the standard deviation and one-eighth in the maximum difference in thickness in the well bottom by itself. 

### 4.3. Compensation of PCR Program

Unlike typical PCR microtubes, the center of the inside and the whole of the outside bottom of the wells have relatively flat surfaces. In addition, the rounded wall allows the cells to collect at the center of the well. Therefore, the cells in the wells were easily found around the center bottom of the wells via microscopic observation. This feature is very useful for single-cell PCR, where the number of cells in the reaction is important. However, it is difficult to perform PCR on a typical thermal cycler for microtubes, because the shape of the well bottom is not suitable for their heating block. Therefore, we chose a commercially available thermal cycler for in situ PCR. The thermal cycler’s heating block has a flat surface. The thermal cycler for in situ PCR is optimized for the PCR of tissues attached onto glass slides [35]. Thus, the thermal profiles of the PCR mixtures applied to this multi-well plate were different from the target temperatures programmed into the thermal cycler, as shown by the green line in Figure 5a. Therefore, the program for PCR cycling was compensated, so that the thermal profile of the PCR mixture in these wells was similar to that in a typical PCR tube. The red line in Figure 5a shows the thermal profile of the mixture in the well by employing the compensated program, which is as follows: 8 s at 99 °C, 1 s at 98 °C, 1 s at 97 °C, and 2 s at 96.5 °C for denaturing; 10 s at 60 °C, 5 s at 61 °C, 5 s at 61.5 °C, and 15 s at 62 °C for annealing and extension. Despite differences in the heating and cooling rates depending on the thermal cycler to be used, the thermal profile for denaturing, annealing and extension in the well was almost approximately that in the tube which was used to perform PCR cycling with a thermal cycler StepOne Plus (Applied Biosystems, Warrington, UK), following the manufacturer’s instructions, with minor modification (the blue line in Figure 5a). By comparing each profile between the PCR mixture and block, even though the rate of temperature elevation of the block was much faster using the conventional method, as shown in Figure 5b, the rate of temperature elevation of the PCR mixture in the well was similar to that in the tube (Figure 5a). This was attributed to the thinness of the well bottom and the low volume of the PCR mixture.

### 4.4. Cell Observation and Counting

After transferring the cells into the PCR mixture, green fluorescence was observed from both KO-expressing (Figure 6a) and non-expressing (Figure 6b) cells, within about 30 min. This may be due to the degradation of TaqMan probes by intracellular DNase. Almost all of the seeded cells were observed around the center bottom of the wells (Appendix A). Many cells were confirmed to have ruptured in the PCR mixture, because the osmotic pressure of the mixture was not adjusted for cells. By distinguishing cells from debris using their fluorescence and bright-field image, we were able to readily count the cells in each well. The number of cells in each well is shown in Figure 7a.

### 4.5. Single-Cell PCR

As the model experiment for single-cell PCR on this 384-well microplate, the amplification and detection of fluorescent protein-expressing DNA was undertaken. PCR was performed using a TaqMan probe and a general master mix (not prepared for single-cell PCR). The DNA amplification was analyzed by measuring the fluorescence intensity of each well, with the image scanner shown in Figure 7b. Air bubbles generated in the wells interfered with the accurate measurement of fluorescence intensity. Therefore, by moving the air bubbles to one half side of the each well by tapping the plate in an upright state, the fluorescence intensity was measured using the other half side of the each well. The PCR results, which are represented as Δ Rn (Figure 7c), were assessed to be linked to the number of cells contained in each well (Figure 7a). On the wells intended to contain single cells (wells C4 to N8 in Figure 7), the genes were amplified in 7/26 (26.9%) wells that actually contained single KO-expressing cells (Table 1). False positives were also confirmed in 1/60 of the negative control wells of the C13 to N17 and in 1/21 wells containing no KO-expressing cells in C4 to N8, which totaled 2/81 (2.5%) wells. This result shows the target genes could be amplified from template DNA in single cells using this newly developed 384-well plate. Next, PCRs from around 10 cells were compared between this approach and conventional PCR methods, using tubes with StepOne Plus (Applied Biosystems). In both methods, DNA amplification was observed in approximately 50% of the 24 samples, and one sample was detected as a false positive (Table 1). Thus, the present method has a sensitivity and specificity comparable to that of the conventional method. We performed PCR amplification from DNA as a simple PCR application, resulting in a relatively low PCR success rate. To increase PCR success rate, reverse transcription-PCR from RNA template present at a higher copy number than DNA is expected to be applied to practical studies.

## 5. Conclusions

The newly developed 384-round-well plate, despite being processed over a large area without a precision mold, was fabricated, with a high accuracy of 13.3 ± 2.3 µm at the thickness of the well bottom. Even if highly precise and expensive molds were used, it would be difficult to achieve such accuracy via the press molding technique, because slight tilting and thermal expansion of the mold become critical problems. In injection molding, it is difficult to fill the thin bottom part of the mold with resin. Using the method of applying thin film as a thin wall, an elongation of the production process, the dissolution and misalignment of the adhesives, and the loss of structural strength of the bonded area are potential problems. Therefore, the present strategy is useful as a microfabrication technique, even without considering the cost issue.

In recent years, along with the increasing attention to single-cell research, many instruments for single-cell analysis have been developed, and are now commercially available. However, their high costs have limited their use. Additionally, these resin products for single-cell research are not compatible with general research equipment, and often require specialized equipment for their use. Therefore, their introduction into actual practice has high hurdles. However, this 384-well plate designed in the format of conventional multi-well plates can be applied to conventional equipment. In this study, we successfully demonstrated the detection of fluorescent protein DNA in single cells as an experimental model for PCR. This plate is expected to be applied to practical studies, such as the analysis of TCR genes in tumor-infiltrating T cells [36,37], and the heterogenetic analysis of cancer [38,39].

In conclusion, the present study shows that elastic PDMS molds can self-compensate for dimensional errors in thickness in corresponding thin-walled parts of resin products. To the best of our knowledge, this report is the first to improve the precision of resin products by deforming a PDMS mold using excessive pressure. By using this mold prepared without precise processing, a 384-round-well plate with bottoms around 10 µm thick was facilely fabricated, with precision beyond that of the mold. Therefore, the proposed molding strategy is promising for facile, low-cost, and higher precision microfabrication.

## Figures and Tables

**Figure 1 micromachines-11-00748-f001:**
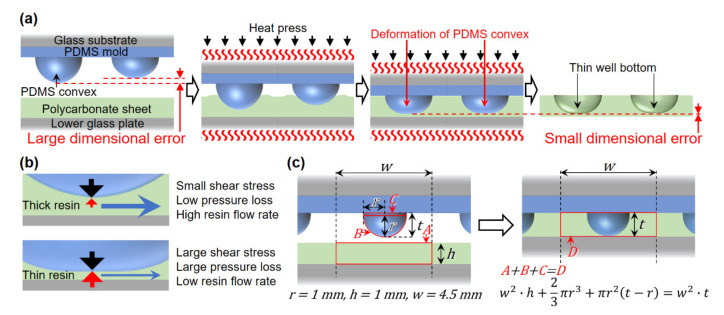
Facile fabrication of thin-bottoms of round wells. (**a**) Scheme for fabrication of the wells by heat-press molding, based on the deformation of a polymethylsiloxane (PDMS) convex. (**b**) Principal for the fabrication of thin well bottoms with a small dimensional error based on the resin flow dynamics. (**c**) Design of the PDMS convex.

**Figure 2 micromachines-11-00748-f002:**
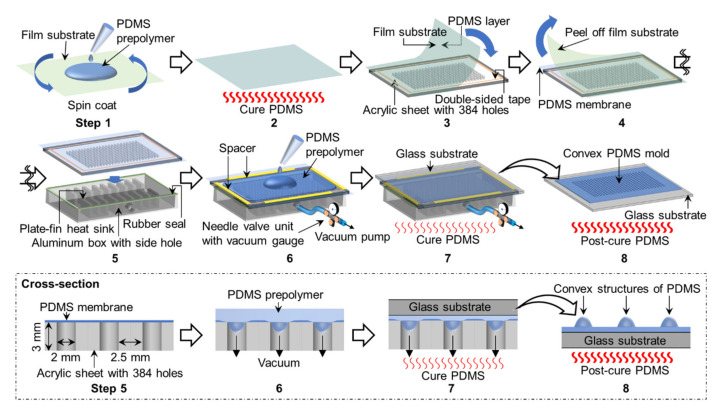
Schematic of the fabrication steps for convex PDMS mold. Steps 1,2: Preparation of a PDMS membrane on a film substrate. Steps 3,4: The PDMS membrane was adhered onto an acrylic sheet with 384 holes. Step 5: The acrylic sheet with the PDMS membrane was assembled onto an aluminum box with a side hole. Steps 6,7: PDMS prepolymer was poured and cured on the concaved PDMS membrane. Step 8: Post-curing the PDMS mold.

**Figure 3 micromachines-11-00748-f003:**
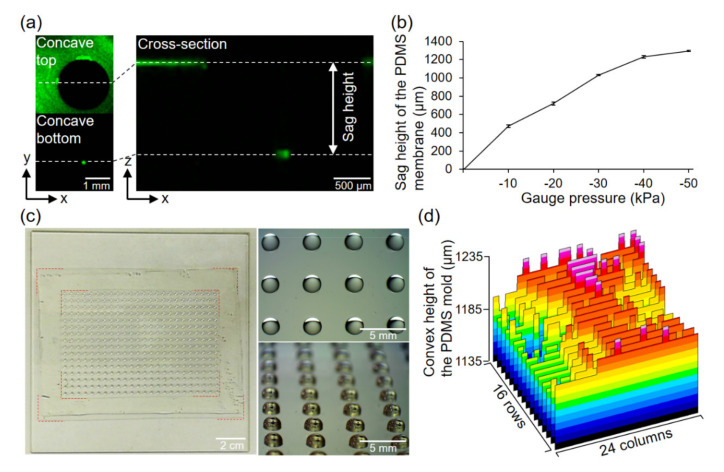
Characterization of PDMS membrane deformation and convex PDMS mold. (**a**) The laser reflection from the surface of the PDMS membrane. (**b**) Sag height of the PDMS membrane at different negative gauge pressures (*n* = 4). (**c**) Left: Photograph of the entire PDMS mold on the glass substrate. The red-dotted line indicates the edge of the thin silicone frame. Right: Close-up image of the mold from above (upper) and diagonally above (bottom) observed with a stereo microscope (S8 APO, Leica Microsystems Japan, Tokyo, Japan). (**d**) Height profile of all convexes on the mold.

**Figure 4 micromachines-11-00748-f004:**
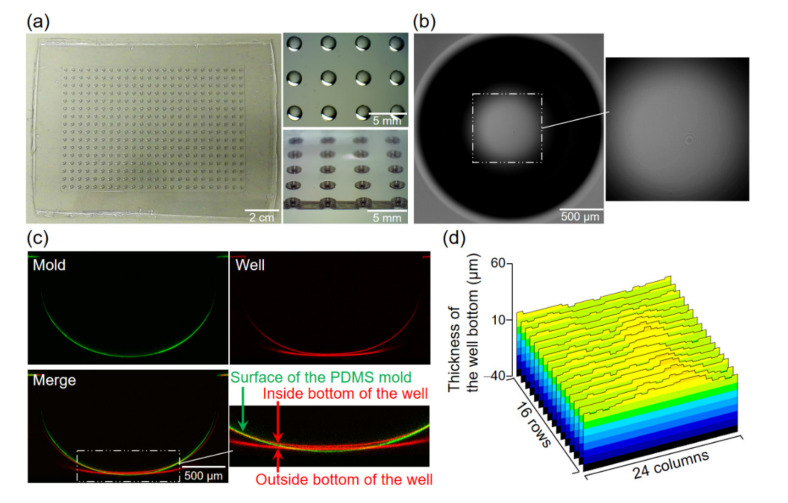
Characterization of the newly developed 384-well plate and comparison with the mold. (**a**) Left: Entire photograph of the 384-well plate. Right: Close-up image of the plate from above (upper) and diagonally above with the cross-section (bottom) observed by stereo microscope. (**b**) Bright-field image of the well bottom. (**c**) Comparison of the cross-section shape between the convex of the mold (pseudo-colored in green) and the well of the plate (pseudo-colored in red), by using confocal z-stack imaging of bovine serum albumin–tetramethylrhodamine (BSA-TAMRA) coated onto their surfaces. (**d**) Thickness profile of the bottom of all wells on the plate. The width of the thickness range was set to 100 µm for comparison with the convex height of PDMS molds (Figure 3d).

**Figure 5 micromachines-11-00748-f005:**
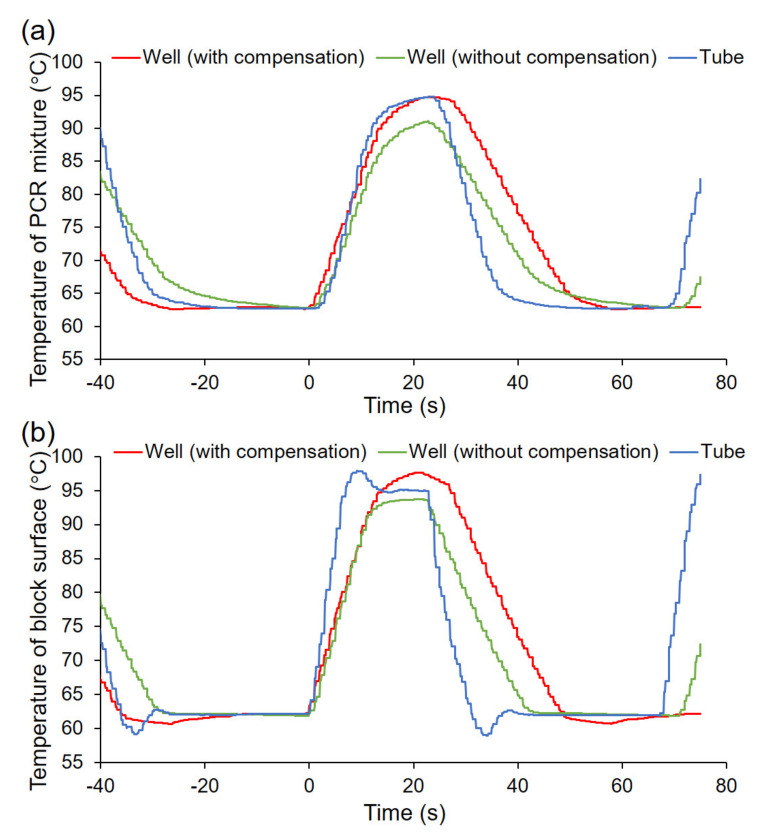
The thermal profiles of the PCR reactions (**a**) in the PCR mixture and (**b**) at the surface of the heating block. The target temperature of the PCR mixture was 95 °C for 10 s for denaturing and 62 °C for 30 s for annealing and extension. The time to start the temperature elevation at the surface of the heating block was defined as 0 s for comparison between the newly developed 384-well plate and a conventional PCR tube.

**Figure 6 micromachines-11-00748-f006:**
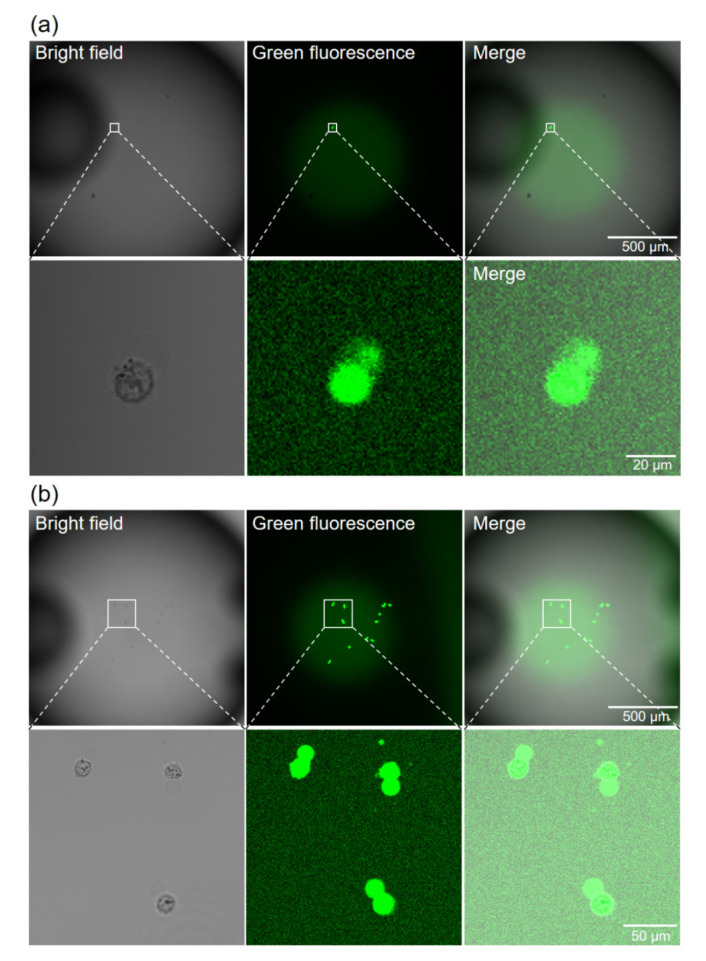
Observation of cells in the PCR mixture on the well bottom. (**a**) Well containing a single KO-expressing cell. (**b**) Well containing 12 non-expressing cells.

**Figure 7 micromachines-11-00748-f007:**
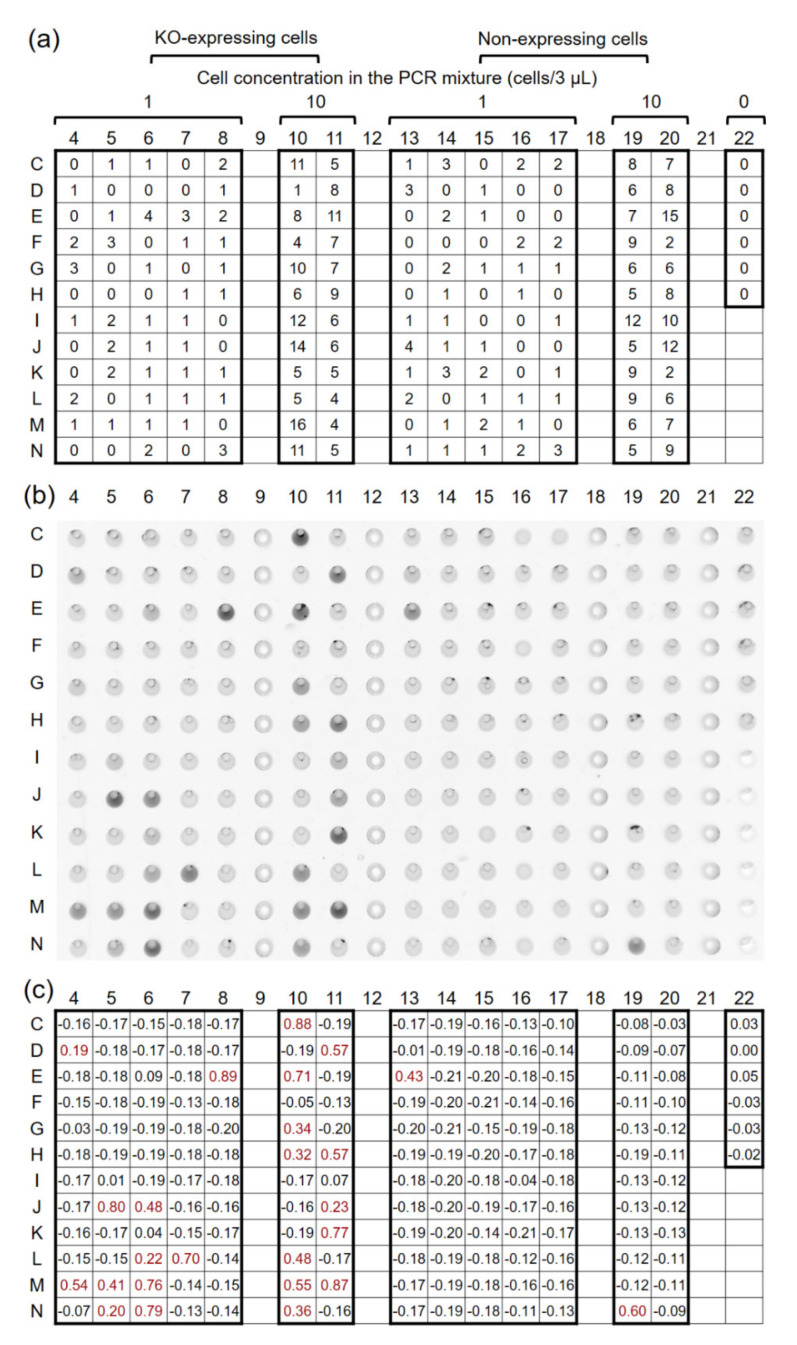
Cell numbers and PCR results for each well. (**a**) Cell numbers and plate layout. (**b**) Plate image of 6-carboxyfluorescein (FAM) fluorescence obtained by an image scanner. (**c**) The values in each well represent ΔRn. Red text indicates the PCR-positive well (ΔRn > 0.1).

**Table 1 micromachines-11-00748-t001:** Summary of the PCR results.

PCR Condition	1 Cell/3 µL in Well			10 Cells/3 µL in Well		10 Cells/20 µL in Tube	
KO expressing	Yes	Yes	No	Yes	No	Yes	No
Cells/well	1	0	0.98 ^1^	7.5 ^1^	7.5 ^1^	*N.A.*	*N.A.*
Wells	26	21	60	24	24	24	24
(False) Positive wells	7	1	1	12	1	13	1
(False) Positive rate (%)	26.9	2.5		50.0	4.2	54.2	4.2

^1^ Average.

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
