# Peer review of "Facile Fabrication of Thin-Bottom Round-Well Plates Using the Deformation of PDMS Molds and Their Application for Single-Cell PCR"

_micromachines, 2020, doi:10.3390/mi11080748_

Round 1

Reviewer 1 Report

This study focus on improving the precision of resin products by deforming a PDMS mold using excessive pressure. By using this mold prepared without precise processing, a 384-round-well plate with bottoms around 10 μm thick was facilely fabricated. It applied in the detection of fluorescent protein DNA in single cells as an experimental model for PCR. The authors show great efforts in the manuscript, where the research design appropriate and the methods described adequately. Minor issues should be considered before publication.

Abstract:

The abstract should more be focusing on improving the precision of the products by deforming a PDMS mold using excessive pressure as a novel processing strategy.

Introduction

The introduction provides sufficient background and includes all relevant references

  • Lines 32-38 References are required for all technologies mentioned
  • Lines 54-59 required Referees that support this phenomenon
  • Lines 61-63 more details should be included about the comparison between the current work and previous works

Materials and Methods

  • Lines 107-108 mentioned 15:1 base to catalyst and Lines 121-122 mentioned 10:1 base to catalyst

why using two different ratios?

  • What is the source of heating and curing in the construction of the PDMS Mold?
  • The polycarbonate product was sonicated for 15 min. What is sonication power and temperature?

Results and Discussion

  • Feedback about contact angle measurement is required
  • The capture of Fig 6 should be rearranged
  • End of this section, more details should be included about the comparison between the previous works and your contribution: e.g deformation method, dimension error, efficiency for PCR,…….

Conclusions

The conclusion should be rephrasing and focuses only on Lines 363-370

Reviewer 2 Report

In this paper, the authors report a fabrication strategy to make thin-bottom round-well plates. Basically, they propose that deforming the PDMS can compensate the dimensional error to obtain uniform resin samples with small fabrication error. The author further demonstrate the applications of the fabricated structures in single-cell PCRs. Overall, this is an interesting fabrication technique, with considerable manufacturing precision. I have some minor comments:

  1. It has been shown that the fabrication technique is superior in term of manufacturing precision and uniformity. However, in the demonstration of single-cell PCR, the authors claimed that the sensitivity and specificity is comparable to that of conventional method. I expect some discussion here, what is the key factor for the sensitivity, how is it connected to the structural parameters of the products?
  2. In the conclusion, I suggest some more discussion of this “universal phenomenon” to guide the readers, e.g., how to make good use of this phenomenon in other manufacturing process based on PDMS.
  3. There are some grammar mistakes throughout the manuscript. The authors should proofread the article carefully.
